# Numerical Analysis of an Inline Metal-Organic Chemical Vapour Deposition Process Based on Sliding-Mesh Modelling

**Xiaosong Zhou [1], Yiyi Wu [2,*], Xiaogang Yang [3] and Chaowen Huang [4]**

[1] School of Mechanical and Electrical Engineering, Guizhou Normal University, Guiyang 550025, China; xiaosong.zhou@gznu.edu.cn

[2] School of Chemical Engineering, Guizhou Minzu University, Guiyang 550025, China

[3] Faculty of Science and Engineering, University of Nottingham Ningbo, Ningbo 315100, China; xiaogang.yang@nottingham.edu.cn

[4] Key Laboratory for Mechanical Behaviour and Microstructure of Materials, Guizhou University, Guiyang 550025, China; cwhuang@gzu.edu.cn

\* Correspondence: yiyiwu@gzmu.edu.cn

**Abstract:** The flow behaviour under the influence of susceptor moving speed is a key factor for the fabrication of high-quality cadmium telluride (CdTe) thin films during the inline metal-organic chemical vapour deposition (MOCVD) process. The main purpose of this paper is to find a method to study the real-time dynamics of transport phenomena inside the reactor. The sliding mesh method is thus proposed and its feasibility is evaluated using computational fluid dynamics (CFD) modelling. A computational grid with 173,400 hexahedral cells is adopted through a grid sensitivity test validation. The simulations show that comparing to 2D modelling, the results of 3D modelling are found to be in good agreement with the experimental data for the temperature range of 628–728 K. Based on the velocity field, the temperature field and distribution of species concentration under different sampling time intervals of 60, 180 and 300 s, the thin film uniformity on both edges of the substrate is found to be influenced by the side effect of the baffle plate. The mass deposited on the substrate is further investigated under different susceptor moving speeds from 0.75 to 2.25 cm/min, and a moving speed between 0.75 to 1.13 cm/min is found to be effectively beneficial to the deposition process.

**Keywords:** CFD simulation; sliding mesh; fluid flows; MOCVD; CdTe thin film

## 1. Introduction

Cadmium telluride (CdTe) has been recognised as one of the most promising low-cost photovoltaic materials for the fabrication of thin-film solar cells [1,2]. The metal-organic chemical vapour deposition (MOCVD) technique is commonly used to prepare CdTe thin film due to its advantages of producing high-structural-quality thin film semiconductor materials over a large area substrate and its great commercial value [3–5]. In the MOCVD process, precursors are introduced into a reactor chamber by a carrier gas, followed by dissociation and/or reaction in the gas-phase near/on the substrate in order to form a stable solid product. By-products are also yielded in the meantime and are carried by main gas flow through the chamber to the exhaust [6].

Many researchers have found that thin film growth rate, uniformity, thickness and microstructure are directly related to solar cell performance, and these parameters are strongly affected by the deposition condition, gas transport process and reactor structure [7–10]. Computational fluid dynamics (CFD) modelling on transport phenomena of complex reactor geometry with the consideration of reaction kinetics has garnered great attention in recent years. For instance, Li et al. [11–13] carried

out a series of studies in three-dimensional model, focusing on the flow and temperature filed, parameter optimization, reaction kinetics model and deposition rate in commercial MOCVD reactor. Tang et al. [14], Li and Xiao [15] analysed the transport phenomena in an industry-scale 96-rod CVD reactor and siemens reactor. Ramadan and Im [16] investigated the film thickness uniformity in a planetary reactor by optimising multiple parameters. Among these commercial reactors, the inline MOCVD reactor has been successfully employed to deposit CdTe thin film by a group of researchers [17–21]; as such, the inline deposition process with a moving substrate is found to be beneficial to thin film uniformity and productivity [22,23]. However, current simulations on the inline process of CdTe growth are mainly conducted in a steady state, while the flow behaviour with the influence of susceptor moving speed still lacks a systematic investigation. In particular, the impact of the interaction between the reactor chamber and substrate is not revealed.

A possible solution to capture the reactor–substrate interaction is to adopt the sliding mesh method. This method has been used in high-speed or rotating turbulence models such as the moving pintle nozzle and multi-impeller mixing tank [24–26]. One of the major advantages of using this method is that the transient deposition behaviour under the moving interaction can be observed at each time step in the simulation, which can further demonstrate the translational motion of the substrate. However, according to our thorough literature review, little has been found on CFD modelling of the inline CdTe thin film deposition process by using this method. This paper is a first attempt to apply this sliding mesh method to establish a full-scale model accordant with the actual inline CdTe deposition process.

One major concern during the application of the sliding mesh method is that it may bring large computation costs due to the massive grid number. In contrast, it has been recognised that inadequate mesh may cause computational inaccuracy and uncertainty of numerical solution if an approximate discretisation scheme has been chosen [27]. In order to balance the numerical accuracy and the computational effort, the mesh refinement should be analysed by a comprehensive consideration of the mesh quality and size. This paper discussed the CdTe thin film deposition through an inline MOCVD deposition process based on CFD modelling. An appropriate mesh was first identified by a grid resolution study, and the suitability of the sliding-mesh method for the CdTe deposition was further examined by comparing the simulation results with the experimental data. The effect of the susceptor moving speed was subsequently investigated, and the transient flow behaviour, temperature field and distribution of species concentration under the reactor–substrate interaction were discussed in details. The simulation results obtained in such way have a practical significance for the future design and optimization of the deposition process.

## 2. Experimental Parameters

An in-line MOCVD reactor which was self-designed in Centre for Solar Energy Research OpTic (St. Asaph, UK) has been employed as a referenced geometric model in CFD simulations. The experimental set-up for inline deposition process of CdTe thin film is illustrated in Figure 1. Dimethylcadmium (DMCd) and diisopropyltelluride (DIPTe) were used as precursors and were further introduced by carrier gas hydrogen ($H_2$) into the reactor. The entire reactor is surrounded by a high-purity nitrogen ($N_2$) containment curtain flow. The susceptor which holds a heated glass substrate is kept at a constant speed, moving from inlet to outlet.

The total flow rate ($F_{total}$) was kept at 0.5 SLM (standard litre per minute) and the flow-rate ratio of DMCd and DIPTe (VI/II ratio) was maintained at 0.55. The substrate temperature ($T_{sub}$) has been varied from 628 to 728 K at a pressure of 950 mbar, and the moving substrate was set at a constant speed ($v_{sub}$) from 0.75 to 2.25 cm/min. CdTe thin film was deposited on the heated substrate with an area ($A_{sub}$) of $0.075 \times 0.05$ m$^2$, and the deposition rate (DR) was calculated using the relationship

$$DR = \frac{m_{deposit}}{A_{sub} \times t_d} \tag{1}$$

where $m_{deposit}$ is the deposited mass of CdTe which is weighted using an electronic analytical balance (Mettler Toledo AB204-N with 0.1 mg precision, Columbus, OH, USA), and $t_d$ is the total deposition time. More detailed information relating to the experimental study and characteristics of CdTe thin film are reported in other published papers [28,29].

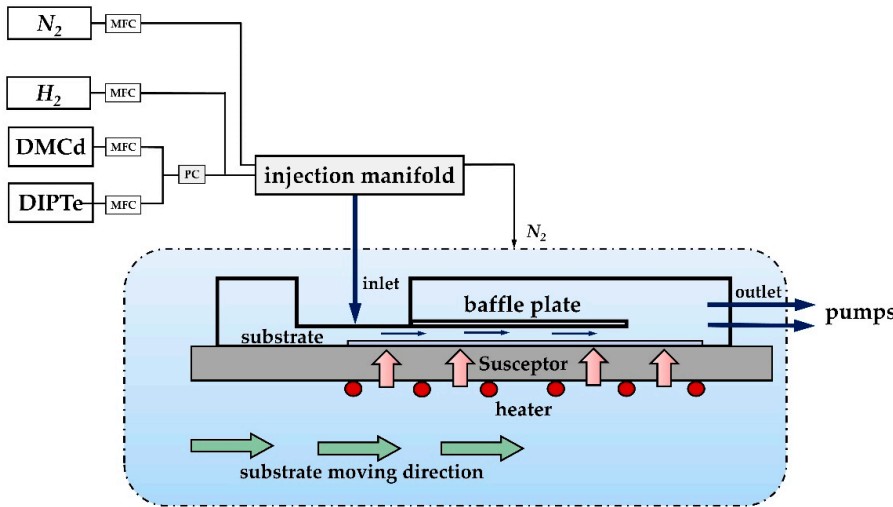

**Figure 1.** Experimental set-up for the inline deposition process of CdTe thin film.

## 3. Numerical Model Description

The schematic diagram of the studied reactor is further shown in Figure 2a. The reactor consists of one vertical injection head, ensuring the precursors can be released normal to the chamber. A baffle plate is located above the substrate to confine the flow path inside the reactor. The main gas flows through the chamber and exhaust from the outlet located on the left.

In order to employ the sliding mesh technique, the entire computational domain is thus divided into two subdomains, where the translational domain (substrate) slides relative to the stationary domain (reactor chamber) along the interface. The mesh interface and the associated two domains are demonstrated graphically in Figure 2b, and between the two computational domains are sliding-mesh interfaces.

For all the numerical models in this study, the gas flow is assumed to obey the ideal gas law, and, accordingly, the gas mixture is calculated by the mixing law. The Reynolds number based on the hydraulic diameter of the reactor falls into a range of 1 to 100 under different conditions, indicating that the gas flow in the reactor can be regarded as laminar flow. According to Yang et al. [23], the estimation of the Reynolds number is mainly calculated by using the physical properties of carrier gas $H_2$, because the concentration of DMCd and DIPTe is very small in comparison to that of $H_2$. In addition, as the density of the carrier gas in the flow is almost unchanged, the flow was considered as incompressible.

The standard conservation equations for mass, momentum and energy are conventionally solved in the domain of reactor chamber [30]. Since the sliding-mesh interface, which connected with the two mesh domains, is capable of dynamically reflecting and updating the mesh motion as a function of time, a modified set of the conservation equations has been adopted at such an interface, which can thus be written as:

$$\frac{\partial \rho}{\partial t} + \nabla \times (\rho \boldsymbol{U} - \rho \boldsymbol{U}_m) = 0, \tag{2}$$

$$\frac{\partial (\rho \boldsymbol{U})}{\partial t} + \nabla \times [\rho \boldsymbol{U}(\boldsymbol{U} - \boldsymbol{U}_m)] = -\nabla p + \nabla \times \boldsymbol{\tau}_{ij} + \rho \boldsymbol{g}, \tag{3}$$

$$c_p \frac{\partial (\rho T)}{\partial t} + c_p \nabla \times [\rho \boldsymbol{U}(\boldsymbol{U} - \boldsymbol{U}_m)T] = \nabla \times (\kappa \nabla T), \tag{4}$$

$$\frac{\partial(\rho\omega_i)}{\partial t} + \nabla \times (\rho \boldsymbol{U}(\boldsymbol{U} - \boldsymbol{U}_m)\omega_i) = -\nabla \times \left(-\rho D\nabla\omega_i - D^T\frac{\nabla T}{T}\right), \tag{5}$$

where $t$ is the time, $\boldsymbol{U}$ is the velocity of gas flow with respect to the stationary reference frame and $\boldsymbol{U}_m$ is the velocity component caused by the motion of moving mesh. The detailed governing equations and the chemistry model were employed in the stationary reactor chamber.

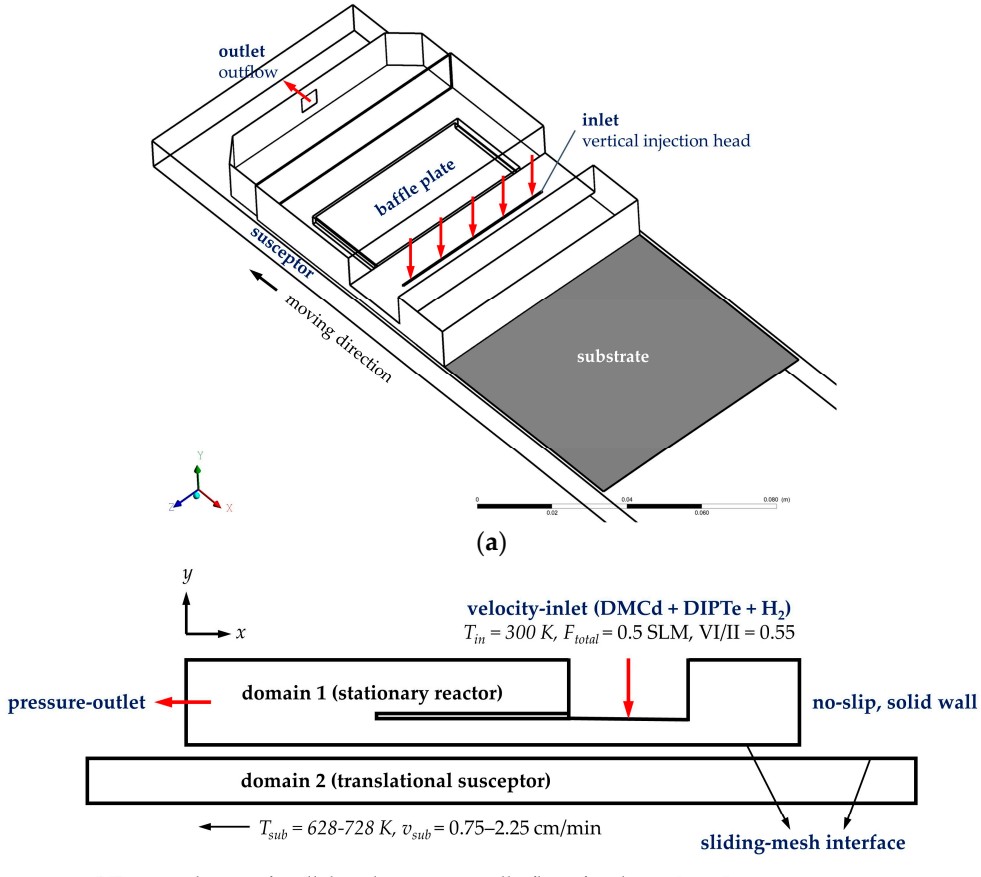

**Figure 2.** Schematic diagram of (**a**) inline MOCVD reactor; (**b**) computational domains and boundary conditions.

The boundary conditions for the reactor chamber domain are: (1) the gas mixture was introduced into the reactor chamber at the inlet-temperature ($T_{in}$) of 300 K; (2) the inlet and outlet were defined as velocity-inlet and pressure-outlet, respectively; (3) no-slip boundary condition was applied for all solid walls; (4) the flux of each species was also assumed to be zero at reactor walls with the exception of substrate wall. Apart from these, the walls between the two domains were defined as interface and the domain located in the side of the translating susceptor was specified by the given temperature with the implement of the overall surface reaction onto the substrate wall. The sliding grid is translating in the negative $x$ direction at a constant moving speed and the sliding grid mechanism takes place at each time step.

The numerical solution of conservation equations was achieved using the commercial code ANSYS Fluent 17.0. The semi-implicit method for the pressure-linked equations (SIMPLE) algorithm was applied for velocity-pressure coupling and the second-order upwind discretisation scheme was employed for the convective terms. The convergence criteria were set at least $10^{-6}$ for energy and $10^{-5}$ for all the other variables in the simulations.

## 4. Computational Grid Resolution

In order to establish a grid-independent model and to determine the favourable grid density for CdTe deposition, the effects of grid structure and resolution on the accuracy of the prediction results are investigated progressively. The domain is divided into a number of non-overlapping finite sized control volumes. Two types of grid are mainly considered, which are tetrahedral volume mesh and hexahedral volume mesh (see Figure 3). The mesh refining process is conducted through a successive increase in the number of grid cells. However, for the hexahedral grid (HG), instead of increasing the number of grids over the whole computation domain, like tetrahedral grid (TG), a non-uniform grid was employed where a greater density grid was placed in the vicinity of the substrate wall and also near the inlet of the showerhead. The specific sizes of elaborate computational meshes of varied cases in the computational domain are listed in Table 1.

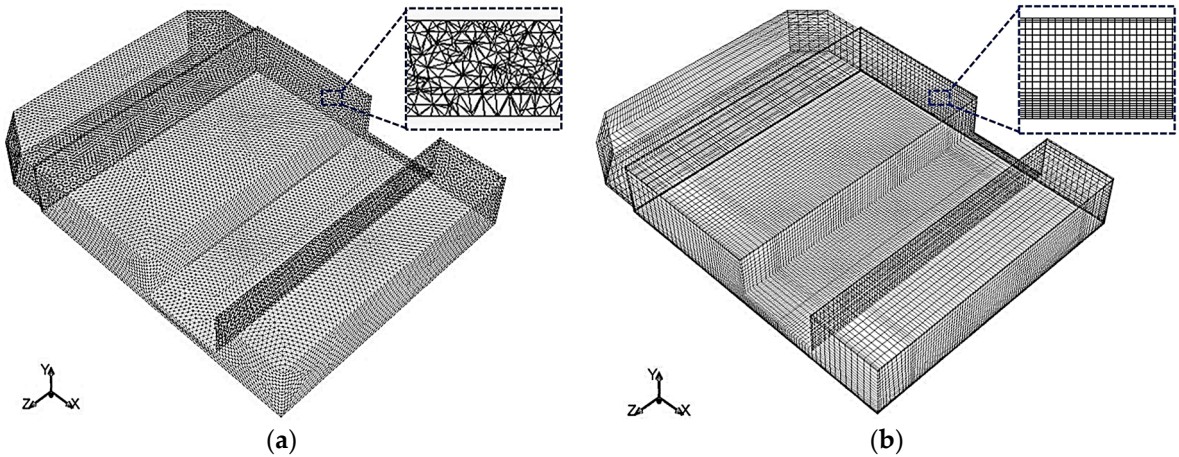

(**a**)            (**b**)

**Figure 3.** Structure of two different grid types: (**a**) tetrahedral grid (TG); (**b**) hexahedral grid (HG) employed in the grid sensitive study.

**Table 1.** Total number of cells for consecutive grid refinement.

| | Tetrahedral Grid | | Hexahedral Grid |
|---|---|---|---|
| **Case** | **Size of the Computational Mesh** | **Case** | **Size of the Computational Mesh** |
| TG 1 | 59,500 | HG 1 | 17,600 |
| TG 2 | 114,600 | HG 2 | 173,400 |
| TG 3 | 268,000 | HG 3 | 746,600 |
| TG 4 | 444,200 | – | – |
| TG 5 | 2,823,700 | – | – |

By a comparison between the simulation results and the experimental data, the predicted CdTe deposition rates with different grids are demonstrated in Figure 4 under the influence of various temperatures. It can be seen that the predicted deposition rate of CdTe for both grid types are generally in accordance with the experiment data (the differences to be within 4% in comparison to the experimental data) and only show a weak sensitivity to the grid refinements. Some discrepancies can be observed for both grid types within the temperature region from 658 to 688 K. This is mainly because (1) the low temperature region (<668 K) is limited by the reactor kinetics, and the predicted growth rate is very sensitive to the deposition temperature, following an exponential form $\propto \exp(-1/T)$. The differences arising in this region are mainly attributed to the consideration of the overall surface chemical reaction on the substrate [22,23,30]; (2) the high temperature region (>688 K) is mainly controlled by mass transfer, and a fast surface reaction is occurred, with the deposition rate being proportional to $T^{3/2}$; (3) between the two regions (668 to 688 K), there exists a transition zone where reaction kinetics and mass transfer both affect the deposition process significantly. The joint effect and

the consideration of the overall reaction both lead to the discrepancies between simulation results and the experimental data.

However, noticeable discrepancies can be discovered between the coarser tetrahedral grid (TG1 and TG2) and the experimental data as presented in Figure 4a, especially in the relatively high temperature region which is dominated by mass transport. This is very likely attributed to the numerical errors arising from the non-orthogonality and coarseness of tetrahedral mesh. The fundamental reason is that the discretisation using tetrahedral mesh relies on the control volume interpolation formulation, where the truncation error is hard to control. Due to the non-orthogonality and irregularity, the series of solutions obtained from the tetrahedral grid simulation show instability of the solution procedure, especially for the coarser mesh (TG1), which fails to converge. When the mesh refines to TG3, the numerical solutions is close to that of hexahedral structured grid (standard deviation ≤ ±1%). Compared with the extraordinary long computing time of the tetrahedral grid (up to a week for TG5), the hexahedral structured grid can greatly save the computational efforts and time (average 85% is saved). Overall, the hexahedral grid performs better than the tetrahedral grid.

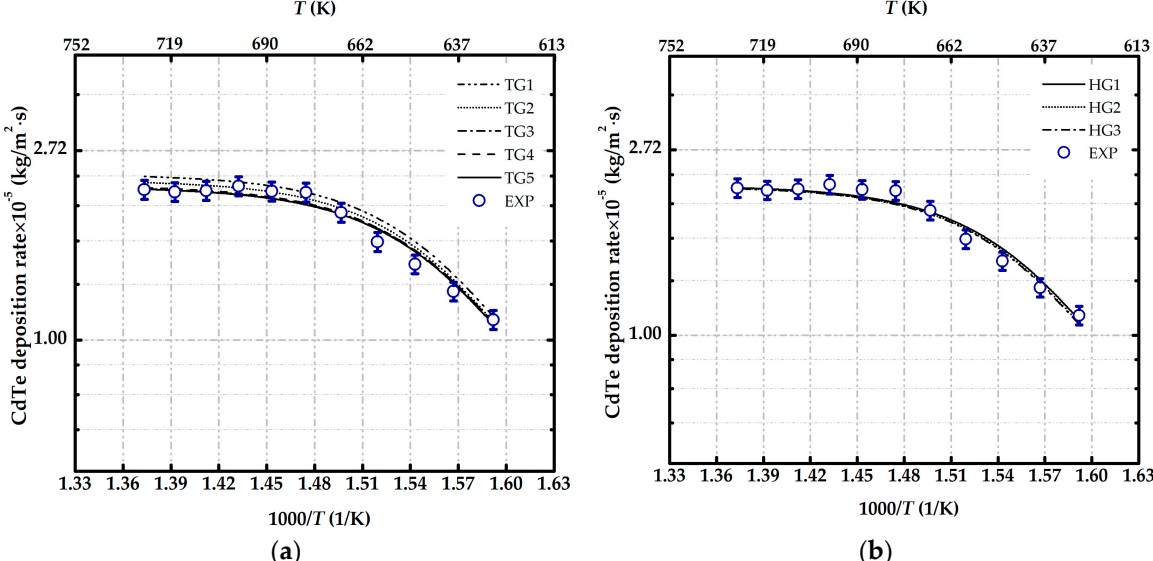

**Figure 4.** Comparisons of experimental data and 3D simulation prediction of CdTe deposition rate by using different types of grid: (**a**) tetrahedral grid (TG); (**b**) hexahedral grid (HG).

For cases of hexahedral grids, as shown in Figure 4b, the disparities of the predicted results of the CdTe deposition rate obtained from HG1 and HG2 are narrow with 0.6% variations, and it is approximately equal (relative standard deviations, RSD = 0.23%) between HG2 and HG3. Refinement in grid failed to show a further improvement in the predicted deposition rate of CdTe, revealing that the predicted values are almost independent of the computational grid. In spite of that, the concentration factor along the $y$-axis plays a dominant role in the deposition process examined (see Figure 5). The concentration in the vicinity of the substrate of HG1 on the substrate surface was shown to be slightly over-estimated due to the limited grid size. The grid of HG2 and HG3 are found to be fine enough to capture both the hydrodynamics and reaction kinetics in the MOCVD reactor. However, between these two cases, HG2 can effectively deduct the expensive simulations in contrast to HG3. Therefore, considering all the factors, hexahedral mesh (HG2) is employed in the remaining work of this study.

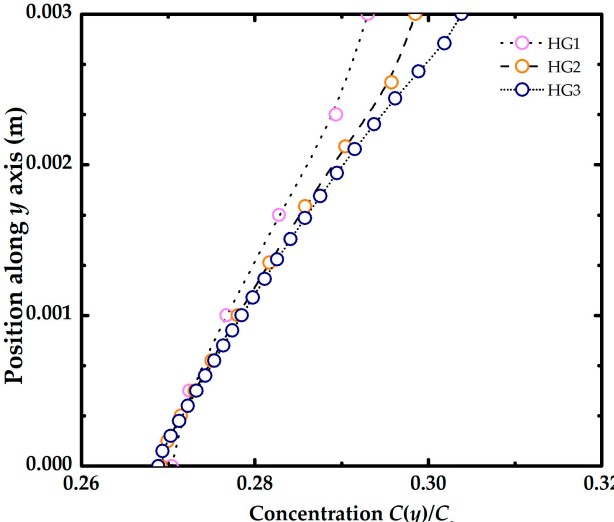

**Figure 5.** Diisopropyltelluride (DIPTe) concentration profiles $C(y)/C_0$ along the $y$-axis at the position of 0.03 m from the susceptor.

## 5. Results and Discussion

### 5.1. Sliding Mesh Technique

A time-dependent modelling of CdTe thin film deposition using the sliding mesh method is conducted with a susceptor moving speed of 1.13 cm/min under the atmospheric pressure. The total gas flow is set to be 0.5 SLM and the VI/II ratio remained at 0.55. The temperature of the substrate is assumed to be uniformly distributed at 668 K. The predicted velocity profiles give out the flow behaviour under different deposition times, as demonstrated in Figure 6. The velocity profiles show little change with time under the baffle plate region, revealing that flow pattern in the deposition zone may not be remarkably influenced by the motion of the susceptor. However, it can be found that the buoyancy-driven circulation loops gradually formed over time on the top of the baffle plate. Such behaviour can be observed more clearly by means of velocity streamlines (see Figure 7). The formation of the recirculation zone is largely related to the temperature distributions. In the upper part, the vertical temperature gradient progressively develops with the influence of the heated-up baffle plate and the cooled reactor wall, further resulting in the enhancement of the recirculation rolls. As can be seen from the mass fraction contour of DIPTe in Figure 7, those rolls pull up the unreacted DIPTe to accumulate in the upper part of the reactor. However, it can be noted that the amount of DIPTe decreases gradually with time increment. This may be attributed to the interactions between the translating susceptor and the reactor chamber. At $t = 60$ s, the susceptor just moves to the location where the deposition on the substrate commences. When the susceptor is gradually moving through the region ($t = 180$ s), the area of the substrate exposed to deposition increases. The chemical species may react immediately when reaching the heated substrate, since a fast chemical surface reaction can be assumed at the substrate temperature of 668 K. It thus leads to a corresponding reduction in the amount of remaining DIPTe as shown in Figure 7b. With the heated susceptor continuously moving, more surface areas of the substrate are in contact with the precursors; thus, more deposition on the substrate surface is achieved. The remaining amount of precursors is significantly reduced, as can be observed from Figure 7c.

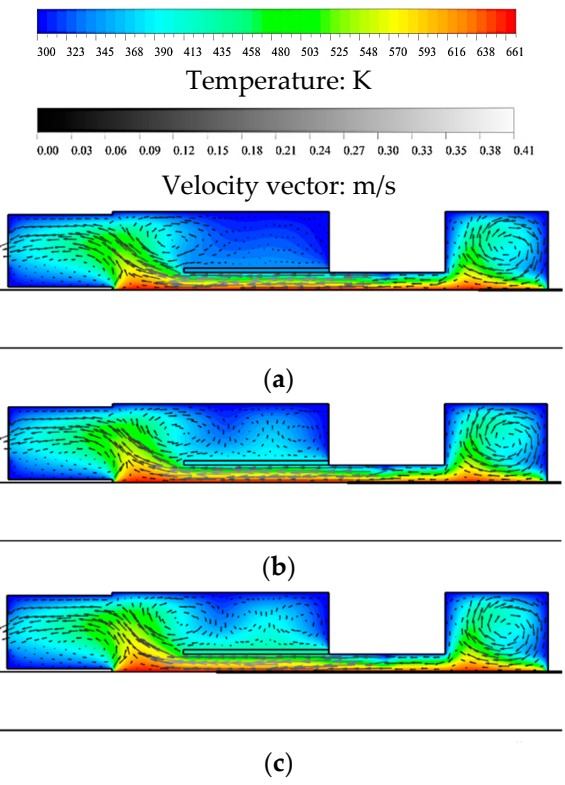

**Figure 6.** Velocity vector coupled with temperature contour of CdTe growth under different deposition times: (**a**) 60 s; (**b**) 180 s; (**c**) 300 s at a temperature of 668 K.

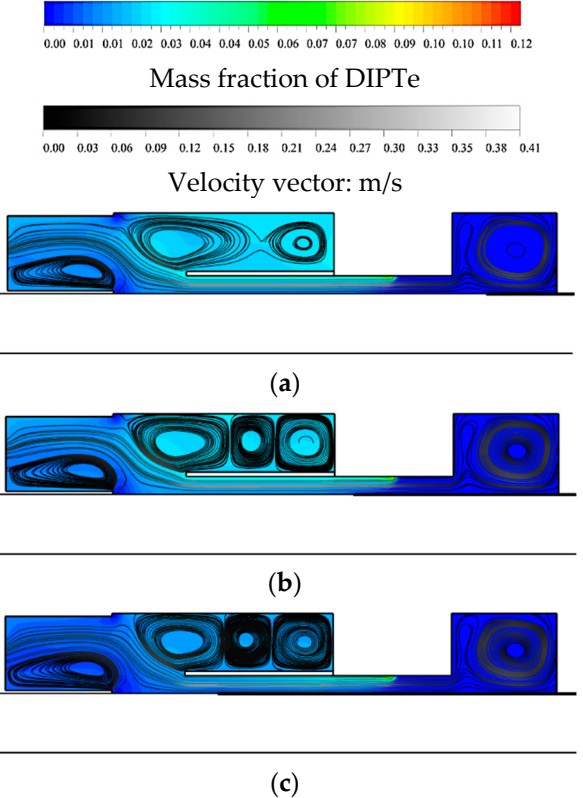

**Figure 7.** Velocity streamline coupled with mass fraction contour of DIPTe for CdTe growth under different time steps: (**a**) 60 s; (**b**) 180 s; (**c**) 300 s at a substrate temperature of 668 K.

The fact of the matter is that a uniform and steady laminar flow in the deposition zone is more desirable in terms of the thin film growth. The enlarged velocity field, as can be seen in Figure 8a, clearly shows that the laminar flow under the baffle plate region tends to be stable very quickly after entering into the reactor chamber. Such behaviour can be further evidenced through a closer look at the detailed velocity distribution (along $y$ direction) at five different $x$ positions: 0.057, 0.047, 0.037, 0.027 and 0.017 m (see Figure 8b). Due to the existence of the large circulating vortex, the stagnation point of the imping jet is skewed to the side of the inlet at $x$ = 0.057 m. However, it can be seen from Figure 8b (2) that the parabolic velocity profile starts to gradually build up, and almost a full parabolic profile is developed at the position of 0.037 m (Figure 8b (3)). This behaviour preserves further downstream at the positions of 0.027 and 0.017 m, as can be observed from Figure 8b (4), (5). It shows that the gas flow has achieved steady status quickly after entering the reactor chamber in the transient simulation; thus, only those velocity profiles at a deposition time of 60 s are depicted.

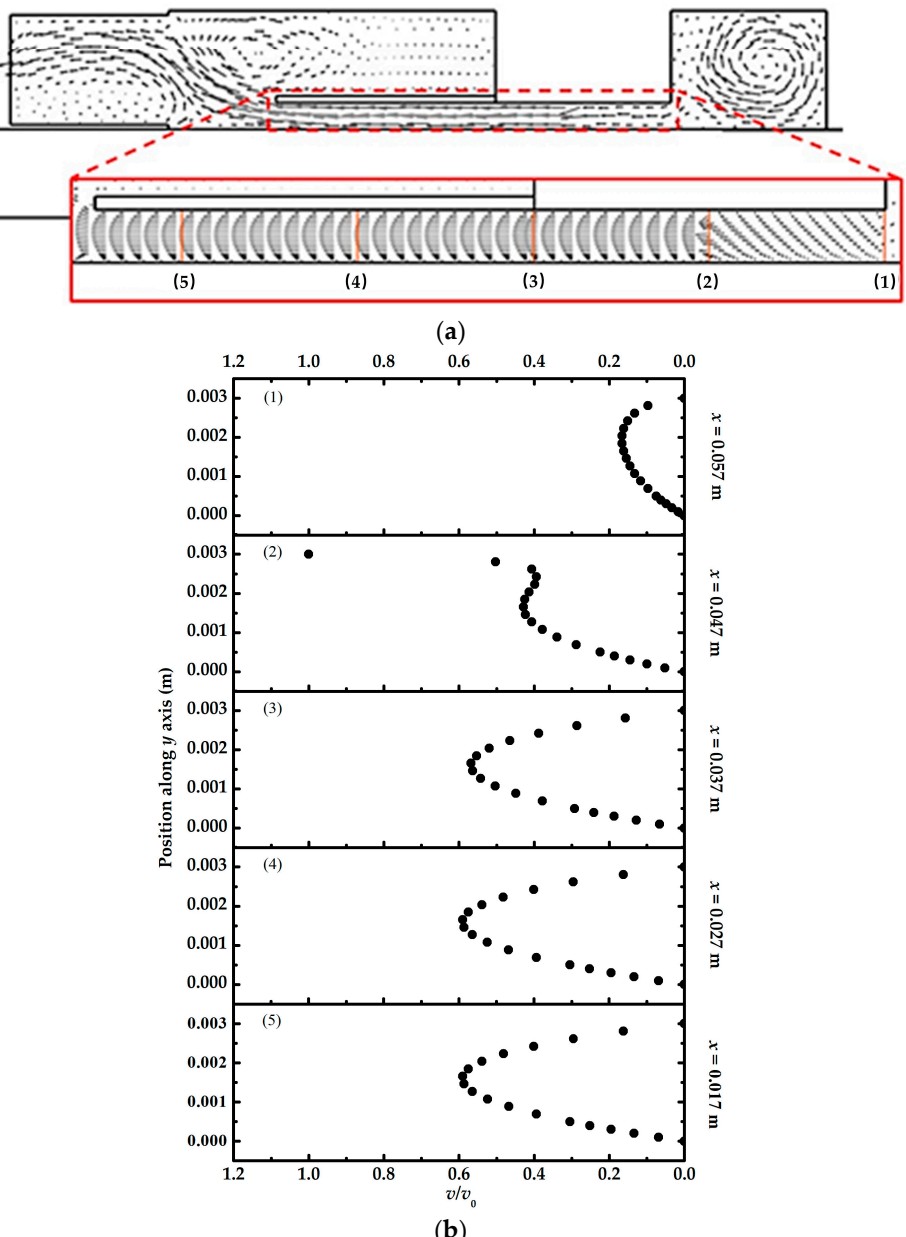

**Figure 8.** Under the conditions of $T$ = 668 K, $t$ = 60 s: (**a**) profiles of enlarged velocity field; (**b**) velocity distribution along $y$-axis at positions of: (**b**) (1) $x$ = 0.057 m; (2) $x$ = 0.047 m; (3) $x$ = 0.037 m; (4) $x$ = 0.027 m; (5) $x$ = 0.017 m.

The thin film uniformity can be further visualised by the contour of CdTe deposition rate on the substrate in Figure 9. As time increases, the deposited CdTe is slowly cumulated on the substrate. It can also be seen that the amount of deposited CdTe is gradually decreased along the substrate as the distribution of the mass fraction of DIPTe reduced in the *x*-axis. The area close to the showerhead has the highest CdTe deposition rate. In addition, due to the recirculation rolls around the baffle plate, the reactive species are gathered on the side of the plate; this might bring species into contact with the heated substrate, leading to a deposition on both edges of the substrate, as illustrated in Figure 9c.

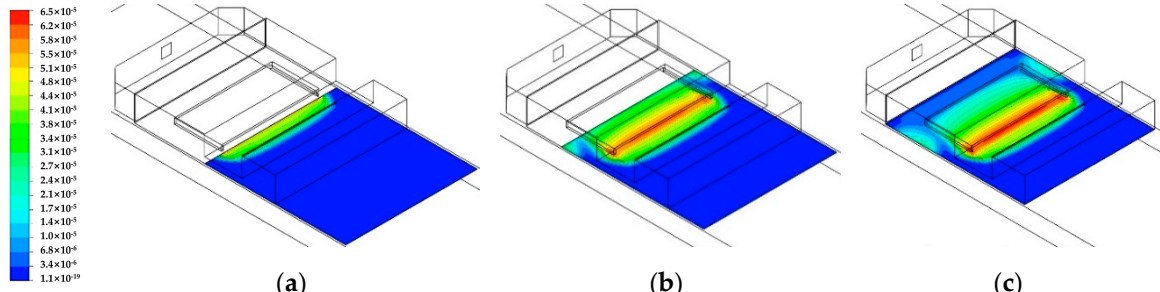

**Figure 9.** Comparisons of the contour of CdTe deposition rate (kg/m$^2$·s) on the substrate with the change over time: (**a**) $t$ = 200 s; (**b**) $t$ = 300 s; (**c**) $t$ = 400 s.

In Figure 10, both 2D and 3D results of CdTe deposition rate using transient simulation are compared with the experimental data under the influence of the substrate temperature. Each data point in the figure is the area-weighted average deposition rate over the substrate.

Generally, both predicted results present a reasonable trend to the experimental data, whereas the 3D simulation result comes to a better agreement with that of the experiment. With regard to 2D simulation, there is an over-prediction in the mass-transport-limited region (from 678 to 728 K), this can be attributed to the ignorance of the side effect of the baffle plate, which may further lead to inaccurate prediction of the temperature field in the chamber, especially in the area above the substrate. In the kinetic-controlled region, the temperature is closely associated with the kinetic term, resulting in the 2D predicted deposition growth rate being lower than the experimental data.

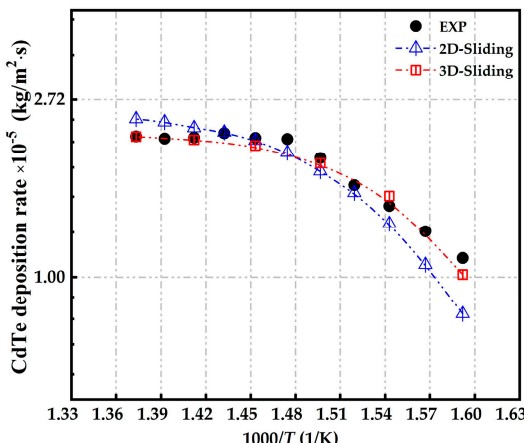

**Figure 10.** Comparisons of the CdTe deposition rate by using the sliding mesh method as a function of the substrate temperature.

## 5.2. Susceptor Travelling Speed

The effect of the susceptor travelling speed is investigated with 3D simulation by adopting different travelling speeds of the susceptor, ranging from 0.75 to 2.25 cm/min. The total flow rate of gas mixture and the VI/II ratio is fixed at 0.5 and 0.55 SLM, respectively. The substrate temperature is

kept at 668 K. The mass of CdTe deposited on the substrate and the corresponding material utilisation as a function of various travelling speeds of the susceptor are shown in Figure 11. The simulation results are generally in good agreement with the experiment data. The mass deposited on the substrate decreases significantly with the increase in the travelling speed of the susceptor, but the decline in material utilisation is modest. This can be interpreted by evaluating the material utilisation ($\eta$ %), which is the ratio of the predicted mass of deposit $m_{predict}$ and the theoretical mass of deposit $m_{theory}$, shown as:

$$\eta \% = \frac{m_{predict}}{m_{theory}} \times 100\%,\tag{6}$$

where $m_{theory}$ is calculated based on the molar supply of the precursors [31], given by:

$$m_{theory} = \rho_{CdTe} N_A \frac{p_{limit} \dot{M}_{limit} t_d}{22.4} \frac{a_0^3}{2},\tag{7}$$

where $\rho_{CdTe}$ is the density of CdTe, $N_A$ is the Avagadro's number, $p_{limit}$ is the partial pressure of the limiting precursor and $\dot{M}_{limit}$ is the mass flow rate of the limiting precursor and $a_0$ is the lattice parameter. Since the other parameters remain unchanged, $m_{theory}$ is simply related to the deposition time $t_d$. According to the previous study [31], the parabolic changing trend of deposited mass ($m_{predict}$) is found very similar to that of the molar supply, which indicates that the kinetic process is almost independent of the substrate moving speed for the studied range. The flow behaviour is further studied under the influence of the susceptor moving speed (Figure 12). It shows that, at the same time ($t$ = 180 s), the exposure area of substrate in the reactor increases with the increasing susceptor moving speed, leading to a slightly different temperature distribution on top of the baffle plate, while the translating susceptor does not have a significant effect on the flow field inside the reactor (see Figure 13). However, the deposition time ($t_d$) becomes shorter with the increase in susceptor moving speed, causing a considerable amount of loss in deposited mass (both in $m_{predict}$ and $m_{theory}$). It can also be seen from Figure 13 that the amount of DIPTe is obviously consumed because of a shorter deposition time of fast-moving substrate (i.e., 2.25 cm/min). As a result, the mass deposited on the substrate decreases with the increase in the susceptor travelling speed, whereas the material utilisation is slightly changed.

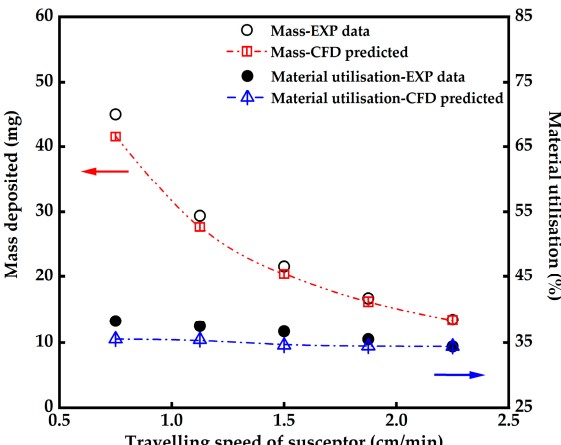

**Figure 11.** CdTe deposited mass and material utilisation as a function of various travelling speeds of the susceptor.

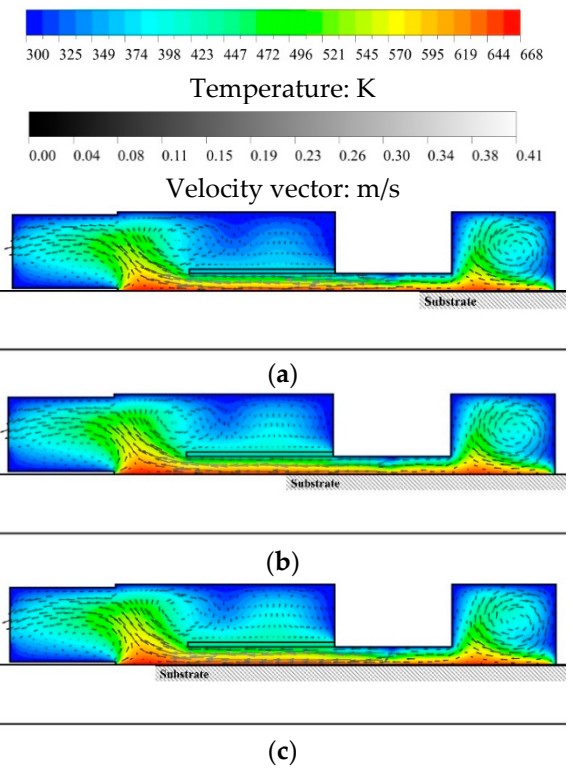

**Figure 12.** Velocity vector coupled with temperature contour of CdTe growth at a time of 180 s with susceptor moving speeds of: (**a**) 0.75 cm/min; (**b**) 1.50 cm/min; (**c**) 2.25 cm/min.

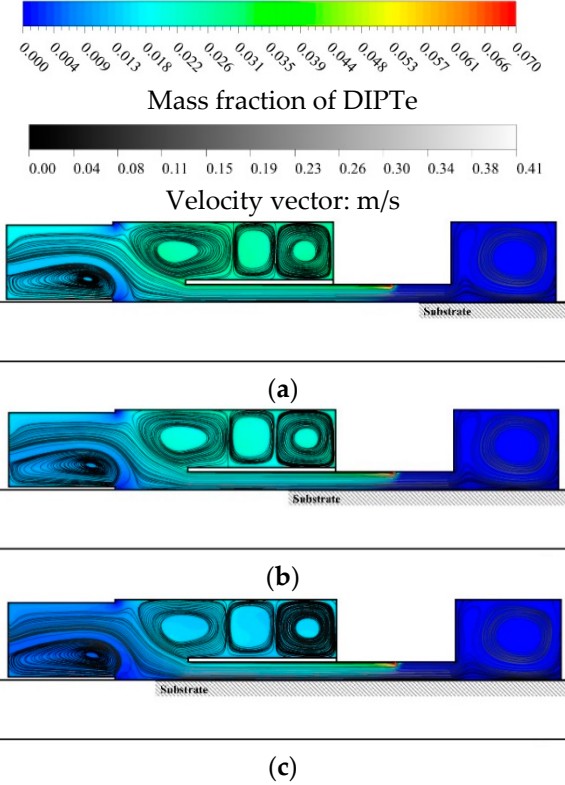

**Figure 13.** Velocity streamline coupled with mass fraction contour of DIPTe for CdTe growth at a time of 180 s with susceptor moving speeds of: (**a**) 0.75 cm/min; (**b**) 1.50 cm/min; (**c**) 2.25 cm/min.

## 6. Conclusions

CFD simulations for an inline MOCVD reactor using the sliding-mesh method have been conducted. The main advantage of this method is that it gives a fresh insight into the real-time dynamics of transport phenomena inside the MOCVD reactor. The main conclusions reached as the results of this study are summarised as below:

- The grid sensitive study shows that the use of hexahedral mesh performs better both in simulation accuracy and convergence speed than the tetrahedral mesh. The number of 173,400 imposed in the stationary domain (reactor chamber) is found to be favourable in this study.
- Both 2D and 3D simulations are performed based on the proposed sliding mesh method, and the results show that 3D modelling fits better with the experimental data, which also demonstrates the feasibility of the sliding mesh method. The use of this method is able to provide a solution which can well interpret the transport phenomena inside the MOCVD reactor from the perspective of transient mass transfer of deposition dynamics. Such time-dependent simulations show that the laminar flow is quickly developed after the gas mixture has been injected into the reactor chamber, which ensures a uniform deposition of CdTe thin film. However, the uniformity on both edges of the substrate are influenced due to the buoyancy-driven recirculation rolls above the baffle plate.
- On basis of the sliding mesh method, the adoption of transient simulations also makes it possible to investigate the influence of susceptor moving speed on the thin film deposition. By varying the susceptor moving speed from 0.75 to 2.25 cm/min, it has been found that the mass deposited on the substrate decreases with the increase in the moving speed, whereas the material utilisation is slightly changed because of a shorter residence time. This finding indicates that a susceptor moving speed between 0.75 and 1.13 cm/min is effectively beneficial in terms of CdTe mass deposition.

**Author Contributions:** Conceptualization and methodology, Y.W., X.Y., and X.Z.; software and calculation, X.Z. and Y.W.; validation, X.Z.; formal analysis, investigation, and resources, X.Z., Y.W., and C.H.; writing—original draft preparation, X.Z.; writing—review and editing, Y.W., X.Y., and C.H.; supervision, X.Y.; funding acquisition, Y.W. and X.Z. All authors have read and agreed to the published version of the manuscript.

**Funding:** This research was funded by Science and Technology Project of Guizhou Province, Grant Nos. [2020]1Y406, [2019]1227; Natural Science Foundation of Guizhou Minzu University, Grant No. [2018]5778-YB20; and Doctoral Research Fund of Guizhou Normal University, Grant No. 0520106/11904.

**Acknowledgments:** The authors gratefully thank technical supports from the research group in Centre for Solar Energy Research (CSER), OpTIC Centre, St Asaph Business Park, UK.

**Conflicts of Interest:** The authors declare no conflict of interest.

## Nomenclature

| | | |
|---|---|---|
| $a_0$ | Å | lattice parameter |
| $A_{sub}$ | $m^2$ | substrate area |
| $c_p$ | J/kg·K | specific heat capacity at constant pressure |
| $C$ | $mol/m^3$ | concentration |
| $C_0$ | $mol/m^3$ | initial concentration |
| $D$ | $m^2/s$ | mass diffusion coefficient |
| $D^T$ | $m^2/s$ | thermal diffusion coefficient |
| $F_{total}$ | L/min (standard), SLM | total flow rate |
| $g$ | $m/s^2$ | gravitational acceleration |
| $m_{deposit}$ | kg | deposited mass |
| $m_{predict}$ | kg | predicted mass of deposit |
| $m_{theory}$ | kg | theoretical mass of deposit |
| $\dot{M}_{limit}$ | kg/s | mass flow rate of the limiting precursor |
| $N_A$ | – | avagadro's number |
| $p$ | Pa | pressure |
| $p_{limit}$ | Pa | partial pressure of the limiting precursor |

| | | |
|---|---|---|
| t | s | time |
| $t_d$ | s | deposition time |
| T | K | temperature |
| $T_{in}$ | K | inlet temperature |
| $T_{sub}$ | K | substrate temperature |
| U | m/s | velocity of gas flow |
| $U_m$ | m/s | velocity component of the motion of moving mesh |
| $v_{sub}$ | m/s | substrate moving speed |
| η | – | material utilisation |
| κ | W/m·K | thermal conductivity |
| ρ | kg/m$^3$ | density of gas flow |
| $ρ_{CdTe}$ | kg/m$^3$ | density of CdTe |
| $τ_{ij}$ | Pa | viscous stress tensor between species i and j |
| $ω_i$ | – | mass fraction of species i |
| CdTe | – | cadmium telluride |
| CFD | – | computational fluid dynamics |
| DIPTe | – | diisopropyltelluride |
| DMCd | – | dimethylcadmium |
| DR | – | deposition rate |
| HG | – | hexahedral grid |
| MOCVD | – | metal-organic chemical vapour deposition |
| RSD | – | relative standard deviation |
| TG | – | tetrahedral grid |
| SLM | – | standard litre per minute |
| SIMPLE | – | semi-implicit method for pressure-linked equations |

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
