# Peer review of "Numerical Analysis of an Inline Metal-Organic Chemical Vapour Deposition Process Based on Sliding-Mesh Modelling"

_coatings, doi:10.3390/coatings10121198_

Round 1
Reviewer 1 Report
The authors have presented numerical analysis of the inline metal-organic chemical vapour deposition process used in the fabrication of cadmium telluride thin-film using a sliding-mesh model. The paper is interesting and has potential to be a journal paper. However, more details are needed to improve the quality and readability of the paper. The following should be addressed in the revised paper.
1. Avoid abbreviation in the title of the paper. Change the title from "Numerical analysis of inline MOCVD process based on sliding-mesh modelling" to "Numerical analysis of the inline metal-organic chemical vapour deposition process using a sliding-mesh model"
2. Line 14. Change "The flow behaviour with the influence of susceptor travelling speed is a key factor ..." to "The flow behaviour under the influence of the susceptor travelling speed is a key factor ..."
3. Line 15. Change "... high-quality CdTe thin films ..." to "... high-quality cadmium telluride (CdTe) thin films ..."
4. Line 22. Change "A further investigation shows that flow behaviour ..." to "Further investigation shows that the flow behaviour ..."
5. Line 24. Change "... and the mass deposited of CdTe and the material utilisation are strong related ..." to "... and that the mass deposition of CdTe and the material utilisation are strongly related ..."
6. Line 25. Change "A lower susceptor moving speed is more beneficial to the deposition process." to "A lower susceptor moving speed is found to be more beneficial to the deposition process."
7. Line 30. The authors should briefly explain the MOCVD process.
8. The introduction section needs to be improved. The authors can improve the introduction section by including pictures of CdTe thin films, this will put the paper in perspective and will be of interest to the readers.
9. In addition to the schematic of the reactor in figure 1. It will be informative if the authors also show the real experimental setup.
10. Figure 1: Provide a schematic with more information. For example, include the boundary conditions, so that it is clearer how the simulations are setup.
11. Include pictures of the complete computational grid and also close up views of the grid that shows the grid refinement details.
12. There are lot of symbols and abbreviations used in the paper. The authors should provide a nomenclature with all the symbols and abbreviations. It will be very useful to the readers and reviewers.
13. The abstract and conclusion section of the paper can be improved by giving more details.
14. The authors should read the paper and check for typos and grammatical mistakes.
Reviewer 2 Report
Dear Authors, thank you for your contribution to this journal. The manuscript presents interesting results regarding the numerical analysis of MOCVD-based CdTe films deposition using CFD modelling. After reviewing it, I enumerate here some points to be addressed and also questions to be answered.
Questions:
1. Pg 3 l86 “For all the numerical models in this study, the gas mixture flow behaviour could be considered that of carrier gas since the precursors have a relatively small concentration”. During the MOCVD process the vaporized chemicals will react and adsorb in the substrate, whilst being constantly fed through the inlet and pumped through the outlet. This could cause local changes in the precursor/carrier gas ratio, thus in the gas mixture behavior. Which experiments have the authors conducted to make sure that the ‘gas mixture flow behaviour’ is the same of the ‘carrier gas flow behavior’? When no experiments have been conducted, can the authors present references to support this statement?
2. Pg 4, figure 3b “It can be seen that the predicted deposition rate of CdTe for both grid types are generally in accordance with the experiment data (the differences to be within 4% in comparison to the experimental data) and only show a weak sensitivity to the grid refinements. However, noticeable discrepancies can be discovered between the coarser tetrahedral grid (TG1 and TG2) and the experimental data as presented in Figure 3(a), especially in the relatively high temperature region which is dominated by mass transport.” One can also observe discrepancies for both grid types for the data points within the temperature region between ~650 to ~710 K. Why?
3. Pg 7-8, Figure 8 and 9. In Figure 8 the authors present a contour plot of the deposition rate, showing that it varies depending on the substrate location in relation to the baffle plate. In Figure 9, the authors present a single value for such. Simulated values: are the values presented in figure 9 an average of all values obtained in Figure 8, or else? Please explain. For the experimental values: at which position was this measured on the substrate? Is this value also an average? Please explain.
4. Have the authors performed any measurement regarding the quality of the CdTe films deposited within this work? I could locate only deposition rate and mass measurements (‘process quality’ measurements), but no measurement regarding the films properties.
Mandatory revision points:
5. Experimental methods section - Please organize all the details regarding the experiments also within this section, including: reactor information, deposition parameters, measurement parameters for deposition rate, amount of mass deposited, and equipment used for it.
6. Pg 5 l163 “and the VI/II ratio is remained at 0.55”. The authors haven’t introduced the VI/II concept before. Please add this info.
7. Figure 1 and Figure 5. In the text, the authors discuss the results based on the location of the baffle plate, but this is not identified in the schematic drawing of Figure 1. Please include all the relevant points/locations being discussed to the schematic drawing. To avoid confusion, I suggest to split Figure 1 into a) schematic drawing of domains (current image) b) schematic drawing of the reactor where the experiments where conducted with indication of all the relevant points/locations.
8. Pg 6: there is an image without identification, although it belongs to Figure 7. Please add an index ‘a’ to it and modify the caption accordingly indicating that the graphs shown in a-e correspond to the locations indicated in the Figure. Right now, this is not clear and in the pdf the figures part were separated.
Suggested revision points:
- Conclusion – the final conclusion that “A relatively low susceptor moving speed is more beneficial to the inline CdTe deposition process” is very general. As the authors already pointed out during introduction, the key question is the balance between film quality and speed/productivity. Thereby, I kindly suggest that the authors modify the conclusion and include the numbers used in it, e.g. “a susceptor moving speed between x and y is more beneficial in terms of mass deposition”.
Round 2
Reviewer 1 Report
The authors have implemented all the corrections suggested in the review report. I can recommend publication of the paper in the journal in the present form.